# Diverse Immunoregulatory Roles of Oxysterols—The Oxidized Cholesterol Metabolites

**DOI:** 10.3390/metabo10100384

**Published:** 2020-09-28

**Authors:** Chloe Choi, David K. Finlay

**Affiliations:** 1School of Biochemistry and Immunology, Trinity Biomedical Sciences Institute, Trinity College Dublin, Pearse Street 152-160, Dublin 2, Ireland; 2School of Pharmacy and Pharmaceutical Sciences, Trinity Biomedical Sciences Institute, Trinity College Dublin, Pearse Street 152-160, Dublin 2, Ireland

**Keywords:** oxysterols, cholesterol, SREBP, LXR, GPR183, ROR, SERM, Ch25h, Cyp27a1, inflammation, cancer, infection, obesity, autoimmunity, endometriosis, immunometabolism

## Abstract

Intermediates of both cholesterol synthesis and cholesterol metabolism can have diverse roles in the control of cellular processes that go beyond the control of cholesterol homeostasis. For example, oxidized forms of cholesterol, called oxysterols have functions ranging from the control of gene expression, signal transduction and cell migration. This is of particular interest in the context of immunology and immunometabolism where we now know that metabolic processes are key towards shaping the nature of immune responses. Equally, aberrant metabolic processes including altered cholesterol homeostasis contribute to immune dysregulation and dysfunction in pathological situations. This review article brings together our current understanding of how oxysterols affect the control of immune responses in diverse immunological settings.

## 1. Introduction

Cholesterol is a vital component of our cells and our bodies. It is a structural component of cellular membranes, plays a role in regulating intracellular signal transduction and is a precursor for the generation of other important molecules such as bile acids and steroid hormones. Cholesterol can be obtained from our diet or synthesised within individual cells and the control of cholesterol homeostasis is carefully regulated. Indeed, elevated overall cholesterol and products of cholesterol metabolism such as oxysterols have been implicated in infection [1] and several diseases such as atherosclerosis, cancer, neurodegeneration and more [2,3,4,5,6]. It is becoming clear that oxysterols can have important regulatory roles within cells and between cells and how oxysterols control the immune system is the focus of this current review.

## 2. Oxidized Cholesterol-Oxysterols

Cholesterol can be spontaneously or enzymatically oxidized, giving rise to a range of oxygenated derivatives of cholesterol that are collectively called oxysterols. These oxysterols are best characterised in the control of cholesterol homeostasis as readily transportable forms of sterol and where they act in a negative feedback loop to regulate cholesterol synthesis, influx and efflux [7]. Oxysterols can be obtained through the diet, particularly in cholesterol-rich foods, or synthesized in various tissues and cells by distinct cholesterol hydroxylases and/or by auto-oxidation with reactive oxygen species [8,9]. Auto-oxidation of cholesterol occurs primarily on the hydrocarbon rings at position 5, 6 and 7 of cholesterol while hydroxylases add oxygen to various other positions (Figure 1). The most abundant oxysterols formed by auto-oxidation include 7β-hydroxycholesterol (7β-HC) and 7-ketocholesterol (7-KC) and are known for their cytotoxic and pro-apoptotic properties [10,11]. They are also found in high amounts in atherosclerotic plaques [2] and in retinal drusen in patients with cardiovascular diseases and age-related macular degeneration [5]. Certain enzymatically synthesised oxysterol species are enriched in particular organs. For instance, 24(S)-Hydroxycholesterol (24(S)-HC), also known as cerebrosterol, is generated by the enzyme cholesterol 24-hydroxylase (Cyp46a1), which is predominantly expressed in neurons of the brain and retina to remove excess cholesterol [12,13]. 22(R)-Hydroxycholesterol (22(R)-HC) is produced in the adrenal gland as an intermediate of steroid hormone production [14]. Outside of the brain, cholesterol 27-hydroxylase (Cyp27a1) produces 27-Hydroxycholesterol (27-HC), which is the predominant oxysterol found in the circulation. It is an intermediate for bile acid synthesis in the liver, an important route for the excretion of cholesterol from the body. Cyp27a1 is also expressed by diverse tissues and cells outside of the liver [9]. Other oxysterols are known to be associated with certain biological states; 25-Hydroxycholesterol (25-HC) levels are substantially elevated in response to inflammation due to the increased expression of Ch25h, most notably in inflammatory myeloid cells of the immune system [15].

Due to the hydrophobic nature of oxysterols, transportation in the circulation or within cells require transportation carriers such as membrane- or protein-based transport systems. In the circulation, the highest levels of oxysterols are present in low density lipoproteins (LDL) and even higher in oxidised LDLs (oxLDLs), while to a lesser extent in high density lipoproteins (HDL) and very low density lipoproteins (VLDL) [16]. Some studies show that oxysterols can also be transported by albumin [16]. Therefore, oxysterols are taken up by cells as part of the LDL fraction through the LDL receptor. Furthermore, oxLDL enriched with oxysterols can be taken up by macrophages and other cells in the vascular wall through scavenger receptors, such as CD36, scavenger receptor (SR) AI or AII, and lectin-like oxLDL receptor 1 (LOX-1), with great impact on atherosclerosis [17,18]. Within the cell, cholesterol, oxysterols and phospholipids oxysterols are transported between/within organelles (such as the endoplasmic reticulum and Golgi apparatus) by lipid transfer proteins such as oxysterol-binding proteins (OSBPs) and OSBP-related proteins (ORPs) [19]. Oxysterols are eliminated from cells by enzymatic sulfation by sulfotransferase Family 2B member 1 (SULT2B1) [20,21] or actively transported out of the cells involving ATP-binding cassette (ABC) transporter proteins [22].

## 3. Oxysterols and Cholesterol Homeostasis

Oxysterols are important regulators of cholesterol homeostasis as they modulate the activities of two families of transcription factor, the sterol regulatory element binding proteins (SREBP) and the Liver-X Receptors (LXR) (Figure 1). Through the inhibition of SREBP transcription factors, oxysterols provide feedback regulation of genes involved in fatty acid and cholesterol biosynthesis and cholesterol uptake [23]. Conversely, oxysterol-mediated activation of LXR induces a transcriptional program that promotes cholesterol efflux from the cell and fatty acid synthesis [24]. In this way, elevated oxysterols, as a result of high cellular cholesterol levels, will inhibit cholesterol uptake and synthesis while promoting cholesterol efflux and removal [25].

### 3.1. LXR

Certain oxysterol species, including 22(R)-HC, 24(S)-HC, 25-HC and 27-HC, are natural endogenous ligands for LXRs (Figure 2A) [26,27]. The LXR family consists of two isotypes, LXRα (also known as NR1H3) and LXRβ (NR1H2), that together are important regulators of cholesterol homeostasis [28]. Both share extensive sequence homology but differ in tissue distribution. LXRα is highly expressed in the liver, intestine, adipose tissue and cells of the myeloid lineage. In contrast, LXRβ is ubiquitously expressed at low levels in most tissues [29]. They exist as obligate heterodimers with retinoid X receptor (RXR). Direct binding of oxysterol to the LXR binding domain induces conformational change and releases the LXR–RXR complex from co-repressors while promoting binding to LXR response elements located in the promoter region of target genes and eliminating excess cholesterol (Figure 2A) [28]. Activation of LXR will result in the mRNA expression of cholesterol efflux transporters such as ATP-binding cassette transporters, ABCA1 and ABCG1. In addition, LXR activation induces the expression of Apolipoprotein E (ApoE) and Cyp7a1, which are also important for cholesterol clearance [28].

### 3.2. SREBP

As discussed above, oxysterols are activators of the LXR nuclear receptors. They are also inhibitors of sterol regulator binding proteins (SREBP), master regulators of genes involved in the lipogenic processes of fatty acid and cholesterol synthesis. There are three isoforms of SREBPs: SREBP1a, SREBP1c and SREBP2. In general, all three isoforms are ubiquitously expressed; however, the ratio of expression may vary across different tissues. SREBP1a and SREBP1c primarily mediate the expression of genes involved in fatty acid synthesis whilst SREBP2 is more commonly associated with the cholesterol synthesis pathway [30,31,32]. Newly synthesised SREBP transcription factors are inserted into the endoplasmic reticulum (ER) membrane where its COOH-terminal domain is associated with a membrane-bound escort protein called SREBP cleavage-activating protein (SCAP). In the absence of cholesterol, SCAP escorts SREBP from the ER to the Golgi apparatus, where two resident proteases; S1P (site 1 protease) and S2P (site 2 protease) cleave SREBP in two places (Figure 2B). As a result, the NH_2_-terminal domain is released and translocates to the nucleus, where it will bind to sterol response elements located in the promoter region of target genes involved in fatty acid and cholesterol biosynthesis [32]. In addition to cholesterol, the oxysterol 25-HC is a potent inhibitor of SREBP activation. In the ER, 25-HC associates with the ER anchor protein INSIG and promotes the binding of INSIG to SCAP, thus retaining SCAP/SREBP at the ER, preventing activation [25]. While 25-HC is well characterised as an inhibitor of SREBP activation, other oxysterols such as 27-HC, 22(R)-HC, 24(S)-HC and 24-25-Epoxycholesterol are also likely to act as SREBP inhibitors as they also bind to INSIG and promote the interaction with SCAP [25]. Indeed, 27-HC has been confirmed to inhibit the activation of SREBP1 in the liver [33].

## 4. Oxysterols and Immune Function

Oxysterols are not only regulators of cholesterol homeostasis but also have important roles in the control of immune responses. There is increasing evidence of diverse actions of different oxysterol species in a range of immune cells subsets. These actions are mediated through the control of LXR and SREBP signalling but also through additional mechanisms including acting as ligands for another nuclear receptor, the Retinoic acid receptor-related orphan receptors (ROR), and for the cell surface receptors G protein-coupled receptor 183 (GPR183) or CXCR2. Oxysterols can exert their regulatory functions within the cell that they are generated in but there is also abundant evidence that they work in a paracrine fashion, acting upon other immune cells [34,35].

### 4.1. Oxysterol-LXR Regulation of Immune Cells

LXR signalling and its impact on inflammation is well established [36]. As numerous oxysterols are natural endogenous activators of LXR nuclear receptors, we will now discuss their diverse functions in controlling the immune response by altering the function and location of immune cells.

#### 4.1.1. Oxysterol-LXR Regulation of Immune Function

LXRs play an important role in immune regulation and have been studied extensively using their natural ligands, oxysterols, and also synthetic agonists such as GW3965. The importance of LXRs for immune homeostasis is illustrated by the fact that mice deficient in LXRs spontaneously develop a lupus-like autoimmune condition [37]. Therefore, LXRs are important for the maintenance of self tolerance. This is explained through roles for LXRs in the control of a number of immune cell subsets. In macrophages, LXR activation can inhibit the expression pro-inflammatory genes triggered by toll like receptor activation, including the cytokines, IL6, TNF and IL8, by suppressing the activity of the NF-κB and AP-1 transcription factors. LXR-induced ABCA1 expression has been reported to antagonise TLR2, 4 and 9 signalling by interfering with membrane lipid organisation and disrupting the recruitment of its adaptor protein, Myd88 [38]. More recently, it was demonstrated that LXR binds in cis to enhancer regions of inflammatory genes containing LXR binding sites and reduces chromatin accessibility [39]. Furthermore, LXR target Mertk tyrosine kinase is important in facilitating the phagocytosis of apoptotic cells by macrophages [37]. Consistent with this, LXR signalling maintains neutrophil homeostasis. LXR signalling is activated when senescent neutrophils are phagocytosed by macrophages and is required for efficient neutrophil clearance [40]. Other immune-related functions of LXRs include regulating T cell responses. For instance, LXR-deficient mice have cholesterol accumulation in the thymus, accelerated thymic involution and impaired T cell development leading to peripheral lymphopenia [41]. Ligation of LXR during T and B cell activation can limit mitogen-driven proliferation of T and B lymphocytes [21]. In addition, the differentiation of CD4 T cells subsets is affected by LXR activation, where it can promote regulatory T cell differentiation [42] and inhibit the generation of Th1 and Th17 cells. The decreased Th17 differentiation in activated CD4+ T cells in the presence of an LXR agonist involves SREBP1c/Aryl hydrocarbon receptor-mediated inhibition of IL17 expression [43,44]. LXR agonists also impact upon the function of B cells and have been described as inhibiting IgE production in human and murine B cells [45].

#### 4.1.2. Oxysterol-LXR Regulation of Dendritic Cell Migration

Oxysterols can affect the migration of dendritic cells (DCs) through LXR-dependent effects on the expression of chemokine receptors and other migration promoting factors. Additionally, they act as direct chemoattractants (see Section 4.4 below). Villablanca et al., showed that human and mouse tumours produce oxysterols that act through LXR receptors and result in decreased CCR7 expression on DCs and a reduction in DC migratory capacity [46]. Similarly, LXR activation was found to reduce CCR7 expression in human monocyte-derived DC [47]. Meanwhile, another study suggests an opposing view of LXR signalling in DC, showing that LXR promotes dendritic cell chemotaxis through inducing the expression of an ectoenzyme CD38 [48]. CD38 is a multifunctional enzyme that has been implicated in leukocyte trafficking during inflammatory responses; CD38^-/-^ mice mount ineffective innate and adaptive immune responses [49,50]. CD38^-/-^ DC or DC with decreased CD38 expression due to LXR deficiency have reduced CCR7-directed migration due to a defect in MAPK signal transduction [48]. Therefore, oxysterol-LXR signalling can clearly affect DC migration, though the exact mechanistic details are not precisely obvious.

### 4.2. Oxysterol-SREBP Regulation of Immune Cells

SREBP is involved in the regulation of immune cell metabolism and function through its control of fatty acid and cholesterol synthesis, the intermediates of which are linked to immunoregulatory signalling pathways.

#### 4.2.1. Oxysterol-SREBP Signalling and Macrophages

SREBP1a activity promotes the inflammatory functions of macrophages, including the production of IL1β, because SREBP1a is required for the expression of Nlrp1a, a component of the inflammasome. Mice that are deficient for SREBP1a show reduced inflammatory macrophages and are resistant to LPS-induced sepsis [51]. The Nlrp3 inflammasome is regulated by the SCAP–SREBP2 complex that forms a ternary complex with Nlrp3 to facilitate trafficking to the Golgi apparatus and efficient inflammasome assembly [52]. Considering that 25-HC prevents SCAP translocation to the Golgi, it would be predicted that 25-HC may also affect the assembly of the Nlrp3 inflammasome. Another study showed that LPS-induced Ch25h expression and the production of 25-HC limits the activation of the AIM2-dependent inflammasome, a process involving reduced cholesterol biosynthesis [53]. Apart from regulating the inflammasome, SREBP signalling also affects phagocytosis and macrophage-signalling pathways. SREBP1a deficient macrophages show reduced phagocytosis as a result of decreased fatty acid synthesis and alterations in plasma membrane leading to reduced interactions between lipid rafts and the actin cytoskeleton [54]. SREBP is important for the anti-viral functions of macrophages through the regulation of cholesterol synthesis and so type I interferon signalling [34,55]. Genetically disrupting SREBP activation through deleting SCAP specifically in macrophages decreased the synthesis of cholesterol in the ER and resulted in the activation of STING/TBK1 signalling to induce IRF3 activity and the expression of interferon regulated genes [56]. In human macrophages, reduced SREBP2 activity was associated with reduced production of the chemokine CCL17 [57]. Aberrant activation of SREBP signalling has been linked to impaired macrophage responses against *Leichmania donovani* parasites due to increased fatty acid synthesis and accumulation [58]. Therefore, there is abundant evidence that oxysterol-SREBP signalling is important for macrophage responses but whether this signalling axis also has a role in the control of other myeloid lineages remains to be determined.

#### 4.2.2. Oxysterol-SREBP Signalling and Lymphocytes

There is evidence that oxysterol-SREBP signalling is an important regulator of IL17-producing lymphocytes. SREBP1c activity can directly regulate IL17 expression in Th17 CD4 T cells [43]. In addition, SREBP-controlled flux through the cholesterol biosynthetic pathway is likely to be important for IL17-producing lymphoctyes because intermediates in this pathway can act as agonists for RORγt, the key transcriptional regulator of these lymphocytes. Indeed, cholesterol biosynthesis has been shown to impact upon the differentiation of Th17 CD4 T cells [59,60]. The activity of SREBP is substantially increased following antigen stimulation of T cells [61]. Deletion of the chaperone protein SCAP, required for SREBP activation, did not affect T cell development but disrupted normal T cell activation. SCAP-deficient CD8 T cells failed to engage in growth and proliferation following TCR activation despite normal TCR signalling [61]. Similarly, in Natural Killer (NK) cells, SREBP activation is required for cytokine-induced growth and effector function. 25-HC treatment or SCAP deletion prevented blastogenesis in response to IL2 plus IL12 cytokine and these SREBP-deficient NK cells lost their ability to kill tumour target cells [62]. Interestingly, in NK cells SREBP activity is crucial for normal catabolic responses. NK cells metabolise glucose through glycolysis coupled to oxidative phosphorylation through a SREBP-dependent citrate-malate shuttle rather than the TCA cycle [62]. Two components of this citrate malate shuttle, ATP citrate lyase (ACLY) and Slc25a1 are SREBP target genes. NK cells that cannot activate SREBP show profoundly reduced glucose metabolism and impaired effector function [62]. Indeed, the defective responses in SCAP^−/−^ T cells were also associated with impaired glucose metabolism, though the underlying mechanism involved was not investigated in this particular study [61].

### 4.3. Oxysterol-ROR Regulation of Immune Cells

Certain oxysterols have also been identified as Retinoic acid receptor-related orphan receptors (ROR) ligands. Like the LXR receptors, ROR family receptors belong to the nuclear receptor super family and as the ligands for these receptors have been somewhat controversial, they are termed “orphan” receptors. Within the ROR family, there are ROR α, β and γ receptors that, when active, bind to target DNA sequences and promote gene transcription through the recruitment of other factors to the promotor. Interestingly, there is some evidence of crosstalk between the LXR and ROR, with each pathway suppressing the other [63]. For instance, RORα signalling promotes the expression of Cyp7b1 which will act to metabolise LXR agonists, including 27-HC [63,64]. Oxysterols have been described as binding to the α and γ isoforms but not the β isoform. Depending on the oxysterol species and the ROR isoform involved, oxysterol binding can either inhibit or activate ROR-dependent transcription, acting as either inverse agonists or agonists (Figure 2C). Oxysterols such as 24-HC and 7α/β-HC are inverse agonists for both RORα and RORγ and will inhibit the transcriptional activity of these nuclear receptors [65,66]. Both RORα and RORγ receptors play a role in the control of immune cell function and in particular in the control of cytokine production. RORα and RORγ have important roles in the development and function of both adaptive and innate lymphocytes [67]. However, whether oxysterols acting as inverse agonists of these receptors have an important role in altering immune responses is not yet clear. Meanwhile, oxysterol species including 22(R)-HC, 25-HC, 27-HC and 7β-27-HC, act as agonists for RORγ nuclear receptors, in particular the RORγt isoform, and so impact upon lymphocyte responses [68,69]. RORγt isoform is of particular interest to immunologists as it is a key determinant of IL17-producing immune cells, including CD4 Th17 and CD27- γδ T cells [70]. Given the role of IL17-producing lymphocytes in various pathological situations from autoimmunity to cancer, there has been a lot of interest in the role played by oxysterols and RORγt. Indeed, it has been shown that mice deficient in Cyp27A1, a key enzyme in generating 7β, 27-HC, have decreased formation of both CD4 Th17 T cells and IL17-producing γδ T cells [68]. The importance of cholesterol in RORγt signalling is further emphasised because intermediates in the cholesterol biosynthetic pathway such as desmosterol are also ligands for RORγt [59,60]. Therefore, both precursors in the pathway towards de novo cholesterol synthesis and oxidised metabolites of cholesterol have a role in RORγt signalling and the generation of IL17-producing T cells.

### 4.4. Oxysterols as Direct Chemoattractants

#### 4.4.1. 7α,25-dihydroxycholesterol and GPR183

Although oxysterols serve as activators of LXR and suppressors of SREBP, subsequent studies have indicated that oxysterols also function as ligands for G protein-coupled receptor 183 (GPR183; also known as EB12) (Figure 3A). 7α,25-dihydroxycholesterol (7α,25-HC) is known as an intermediate of bile acid synthesis and is unable to activate LXR, but is also a potent natural chemoattractant for immune cells expressing GPR183 [71]. 7α,25-HC is synthesised in a stepwise manner by Ch25h and Cyp7b1 and is highly expressed by stromal cells at the perimeter of the follicles in lymphoid organs [72]. In contrast, the enzyme 3β-hydroxy-delta-5-steroid dehydrogenase (HSD3B7), which degrades 7α,25-HC, is highly expressed in the T cell zone by DCs and stromal cells in this area. Therefore, differential expression of these genes in distinct compartments of the lymphoid organs generates a gradient of 7α,25-HC. Interestingly, 25-HC, which differs by only a single hydroxyl group, was inactive as a ligand for GPR183.

It was illustrated that GPR183 is crucial for the migration and positioning of B cells in lymphoid tissues to mediate T cell-dependent antibody responses. GPR183 is rapidly upregulated in antigen activated B cells, which will migrate towards the outer follicular regions with high density of 7α,25-HC within 2 h. After 6 h, together with increased CCR7 expression, B cells migrate towards the T cell zone to seek T helper cells. Two to three days after immunisation, B cells migrate back towards the outer follicular regions by downregulating CCR7 while maintaining GPR183 expression. After a week of immunisation, B cells begin to proliferate and differentiate to germinal centre B cells where they will downregulate GPR183 expression to migrate toward the centre of the follicle where 7α,25-HC concentrations are low. Alternatively, some maintain GPR183 expression and migrate towards the marginal zone bridging channels to differentiate into plasma cells. GPR183-deficient B cells were unable to migrate to extrafollicular regions and remained in the central follicular areas, resulting in defective plasma cell response. [73,74,75,76]. Additionally, splenic CD4+ DCs require GPR183 expression for correctly locating to the marginal zone bridging channels in the spleen. This was shown to be important for maintaining their homeostasis and capturing blood-borne antigen for mounting appropriate CD4+ T cell activation and T cell-dependent antibody response [77,78]. In GPR183-deficient mice, there was lower frequency of CD4+ DCs in the spleen and lymph node as CD4+ DC failed to localise to the marginal zone bridging channels where B cells provide trophic factor to support DC survival. T cells are mostly retained in the T cell zone due to the high expression of HSD3B7. However, T cell activation induces GPR183 expression as early as 12 h, especially in T follicular helper cells (Tfh), an important subset for mounting antibody response [79]. It was demonstrated that early GPR183 expression enables T cells to migrate towards the outer T cell zone where they encounter CD4+ DCs, which promote Tfh differentiation. After two weeks of immunisation, Tfh downregulate GPR183 and migrate towards the centre of the follicle residing in the germinal centres. T cells deficient in GPR183 have reduced Tfh numbers and have defective humoral responses [80]. More recently, ILC3 but not NK cells were found to express GPR183 and this was crucial for ILC3 localisation to the lymphoid structures such as cryopatches and isolated lymphoid follicles in the colon and for regulating intestinal homeostasis [81]. Furthermore, GPR183 expression in ILC3 is important for its distribution in mesenteric lymph nodes and their antimicrobial functions as GPR183-deficient mice were more susceptible to *C. rodentium* infection compared to WT mice [82]. GPR183-dependent localisation in the mesenteric lymph node, specifically within the interfollicular regions, is critical in fine-tuning Tfh-dependent B cell responses and mucosal IgA levels towards both commensal and pathogenic bacteria in the intestine [83]. Collectively, these data show the importance of 7α,25-HC in maintaining the positioning of immune cells expressing GPR183 in steady state in order to allow an effective immune response.

#### 4.4.2. 22(R)-HC and CXCR2

Neutrophils are traditionally known for their antibacterial function but it is becoming clear that tumour-associated neutrophils (TAN) and their myeloid precursors play a significant role in cancer biology [84,85]. When neutrophils traffic into tumours, they are referred to as TAN and in mice, TAN can be defined as CD11b^high^Gr1^high^ with low expression of macrophage markers [85]. Their role in cancer remains controversial and context-dependent; like macrophages, TAN can display plasticity and polarise towards antitumour (N1) or protumour (N2) phenotype depending on their environmental stimuli [86,87]. However, enhanced presence of TANs within tumour tissue often correlate to poor patient outcomes but it is unclear whether TANs directly contribute to cancer progression. Of note, tumour cells express a range of chemokines including CXCL2, CXCL5 and others [84] that facilitate recruitment of neutrophils to the TME through CXCR1 and CXCR2, and high levels of CXCL5 are expressed in a murine model of hepatocellular carcinoma (HCC) and correlate with poor prognosis in human HCC patients [88]. Mouse T cell lymphoma and Lewis lung carcinoma cells express a number of oxysterol species such as 24(S)-HC, 25-HC, 27-HC, 19-HC and 22(R)-HC, which have been shown to recruit tumour-inducing CD11b^high^Gr1^high^ myeloid cells (including neutrophils). It was illustrated that 22(R)-HC is a naturally-occurring chemotactic ligand of CXCR2 and facilitated tumour growth by promoting neoangiogenesis or immunosuppression (Figure 3B). This recruitment activity was found to be independent of the LXR pathway as LXRα, β and αβ deficient neutrophils were able to migrate towards 22(R)-HC embedded matrigel plugs [89].

### 4.5. Immune Cell-Derived Oxysterols as Estrogen Receptor Modulators

Oxysterols are also identified as endogenous selective estrogen receptor modulators (SERM) that have both agonist and antagonist activities that are cell type-specific (Figure 2D) [90]. This is another mechanism through which the immune cell-derived oxysterols can impact upon disease states including atherosclerosis and cancer. For instance, 27-HC along with its synthetic enzyme, Cyp27a1 are abundantly expressed in atherosclerotic lesions, in both endothelium and macrophages. 22(R)-HC, 24(S)-HC, 25-HC and 27-HC were able to inhibit 17β-estradiol (E_2_) activation of estrogen receptor (ER) α and β, with the most potent inhibitor being 27-HC [91]. Elevated 27-HC concentrations promoted atherosclerosis in APOE^−/−^ Cyp7b1^−/−^ mice. It was reported to inhibit the protective cardiovascular properties of estrogen by reducing mRNA expression of nitric oxide synthases (iNOS and eNOS) that are important for vasodilation and endothelialisation after vascular injury [91]. In addition to its regulation of nitric oxide production, 27-HC also increases macrophage infiltration and promotes pro-inflammatory mediators, including IL6, MMP-9 and TNFα in an ER-dependent and LXR-independent manner [92]. Multiple human ER+ breast cancer cell lines, breast tumour samples and tumour-associated macrophages have elevated expression of Cyp27a1. 27-HC partial ER agonist activity was sufficient to promote ER+ tumour growth, independent of the LXR pathway, in an autonomous manner [93,94]. 27-HC also induces MDM2-mediated p53 inactivation in ER+ and but not ER- breast cancer cell lines, thus facilitating cell proliferation [95]. Along with estrogen activity in the cardiovascular system and tumour growth, estrogen also plays an important role in bone health by maintaining proper bone density. It was found that elevated 27-HC concentrations in Cyp7a1^-/-^ mice led to decreased bone mineral density as well as trabecular architecture. Interestingly, E_2_ supplementation only partially rescued defects, suggesting there is an additional target [96].

## 5. Oxysterols and Pathological Immune Dysregulation

### 5.1. Cancer

It is becoming increasingly apparent that the ability of tumours to evade or suppress the anti-tumour immune response is a key hallmark of cancer. Indeed, cancer immunotherapy now provides a range of new and robust therapeutic opportunities since the initial discovery of check-point inhibitors, antibodies that can restart the anti-tumour immune response by removing inhibitory signals. The immune evasion strategies utilised by tumours are diverse, ranging from recruiting immunosuppressive immune cells to creating a hostile metabolic environment for infiltrating T cells and NK cells. The inhospitable metabolic conditions within the tumour microenvironment can include hypoxia, low pH levels and a scarcity of nutrients like glucose. There is also evidence of increased levels of cholesterol in tumours and tumour-infiltrating immune cells [97,98,99]. A recent study by Ma et al. using the B16 mouse model of melanoma showed that infiltrating CD8 cytotoxic T cells were loaded with high levels of cholesterol that lead to an anergic phenotype [99]. Oxidised cholesterol metabolites can also be generated in the tumour microenvironment and represent another mechanism of immune evasion. Oxysterol production by tumours enrich the tumour microenvironment (TME) with immunosuppressive immune cells and inhibit antitumour immune cells, thereby promoting self-growth [3]. For instance, 22(R)-HC recruits immunosuppressive neutrophils to the TME and promotes angiogenesis in a CXCR2-dependent manner (Figure 3B) [89]. 27-HC has also been linked to the recruitment of tumour-promoting immune cells to the TME. 27-HC depletes CD8 T cells by attracting neutrophils and γδ T cells and so contributes to tumour metastasis, though the exact mechanisms involved have not been elucidated [100]. 25-HC also attracts tumour-promoting immune cells such as tumour-associated macrophages/monocytes (TAM), to the TME [101]. Furthermore, oxysterols have an indirect effect on DC migration through downregulating CCR7 expression, interfering with DC migration to the lymph nodes and so inhibiting DC-mediated antigen presentation and the induction of anti-tumour T cell responses (Figure 3B) [46]. Oxysterols, as inhibitors of SREBP, have been shown to be potent inhibitors of NK cell cytotoxicity against tumour cells [62]. Taken together, it is clear that promoting the production of oxysterols is an immune evasion strategy that can be used by tumours to promote growth and metastasis.

The direct effects of oxysterols on cancer cells have been extensively studied. Depending on the cancer context, oxysterols have positive and negative impact on tumour cell proliferation and metastasis. For instance, in breast cancer models, 27-HC binding to estrogen receptor in breast carcinoma MCF-7 causes a unique conformational change that is not seen with estrogen or other SERMs. This unique change is followed by transcription of genes involved in tumour growth [90,93]. Furthermore, 27-HC promotes epithelial to mesenchymal transition with increased expression of vimentin and decreased expression of e-cadherin, suggesting a role in promoting breast cancer migration and metastasis [102,103]. Conversely, there have been studies suggesting negative effects on tumour growth. Of note, downregulation of SREBP expression was reported to induce ER-stress, and accumulate reactive oxygen species and apoptosis in breast cancer and glioblastoma cells [104]. 25-HC inhibition of SREBP decreased cell proliferation of colon cancer cells and knockdown of SREBP inhibited xenograft tumour growth in vivo [105]. Moreover, accumulated levels of 27-HC were found in human gastric tumour tissues and were reported to decrease gastric cancer cell proliferation and migration by modulating LXR signalling [106]. Similarly, 27-HC was reported to inhibit colorectal cancer cell proliferation independently of ER or LXR signalling but rather through decreasing Akt activation [107]. Collectively, the direct actions of oxysterols upon cancer cells are varied and need further investigation.

What cells within the TME make oxysterols? An interrogation of some publicly available datasets reveals that the enzymes Ch25h and Cyp27a1 are expressed in a range of cells within tumours. Most notably, they are expressed by myeloid cells including macrophages and dendritic cells, but also non-immune cells such as fibroblasts [108,109]. Tumour cells themselves can also express both Ch25h and Cyp27a1, though this is variable and oxysterol production seems to be associated with solid tumours more so than for haemopoietic tumours (see www.portals.broadinstitute.org/ccle). The conversion of cholesterol to 22(R)-HC is the first step in steroid hormone synthesis and is carried out by Cyp11a1, which is primarily expressed in the adrenal gland. In humans, Cyp11a1 is not expressed by cancer cell lines or in primary cancer tissue (www.portals.broadinstitute.org/ccle, www.proteinatlas.org) [110]. However, there is evidence that Cyp11a1 is expressed in tumour-infiltrating T cells. This Cyp11a1 expression has pro-tumour functions as T cells’ specific deletion of Cyp11a1 leads to reduced tumour growth, using mouse tumour models [111]. It will be interesting to determine whether Cyp11a1 is expressed in infiltrating T cells in human cancer, if 22(R)-HC is generated in appreciable amounts and, if so, whether this leads to neutrophil recruitment.

Targeting oxysterol-signalling pathways as a cancer immunotherapy is being considered. For instance, pharmacological LXR agonist, RGX-104-001 is in phase I clinical trials in patients with solid tumours and lymphoma. In a murine model of melanoma, administration of RGX-104-001 reduces the survival and abundance of myeloid derived suppressor cells (MDSCs) in vitro and in vivo, thereby enhancing cytotoxic T lymphocytes antitumour activity [112]. However, past studies have implied that LXR activation can both promote and suppress inflammatory responses. Activation has a negative impact on DC antigen presentation and mitogen-driven proliferation of T and B lymphocytes [21,46]. Silencing SREBP activity has been suggested as a potential cancer therapy as emerging evidence shows that SREBP is required to sustain cancer cell proliferation [105,113]. However, SREBP activation is essential for murine NK cell metabolism and cytotoxic activity [62]. In addition, CD8 T cells deficient for SREBP activity fail to engage in growth and proliferation following TCR activation despite normal TCR signalling [61]. Therefore, the effects of LXR ligands or SREBP inhibitors on other aspects of the immune system need to be further characterised and considered in order to avoid potential adverse effects.

### 5.2. Obesity

Obesity is a complex condition involving excessive body fat and is a factor in a range of chronic inflammatory diseases. Obesity causes profound changes in the immune system including chronic inflammation and the dysfunction of immune subsets including NK cells and Mucosal-associated invariant T cells (MAIT) cells [114,115,116]. In addition to elevated levels of lipids and cholesterol, the levels of oxysterols are also elevated in obesity. In various mouse models of obesity, increased levels of oxysterols, including 27-HC and 25-HC, have been measured in a range of tissues [117,118,119,120]. Increased oxysterols have also been measured in the plasma of obese female humans, although interestingly not in males [121,122]. Recently, Ch25h expression in adipose tissue macrophages has been linked to adipose tissue inflammation in humans and mice. Indeed, obese Ch25h^−/−^ mice had reduced diabetes phenotype and were more sensitive to insulin [122]. Another study has linked 27-HC to macrophage function in obese mice. Elevated 27-HC resulted in increased body mass of mice in an ER-dependent manner and this was associated with increased numbers of inflammatory M1 macrophages in the adipose tissue and increased expression of proinflammatory cytokines, IL1β and TNFα [120]. However, there are other contexts where elevated oxysterols in obesity appear to promote anti-inflammatory immune functions. In a mouse model of non-alcoholic fatty liver disease, mice that lacked the expression of Cyp27a1, and so the production of 27-HC, had increased accumulation of cholesterol in Kupffer cells in the liver and increased liver inflammation [123]. In contrast, the administration of 27-HC or the overexpression of Cyp27a1 led to decreased cholesterol accumulation in Kupffer cells and reduced hepatic inflammation [123,124]. It is suggested that endogenously generated 27-HC has anti-inflammatory effects in Kupffer cells through promoting cholesterol trafficking away from the lysosome [124]. Obese mice and humans also show striking NK cell dysfunction both in terms of metabolic pathways and NK cell cytotoxic function [114,125,126]. While this dysfunction was initially attributed to increased fatty acid accumulation in NK cells, it is tempting to speculate that elevated oxysterols may also play a part, especially considering that oxysterol-sensitive SREBP has been identified as a key regulator of NK cell metabolism and function [62].

### 5.3. Chronic Inflammatory Disease

There is evidence that cholesterol and oxysterols contribute to the pathogenesis associated with a number of chronic inflammatory conditions. The best characterised chronic inflammatory disease associated with cholesterol and cholesterol metabolites is atherosclerosis, a condition where there is an accumulation of lipids, cholesterol and other substances in the artery walls leading to restricted blood flow. The formation of atherosclerotic plaques in the arteries of the heart leads to coronary heart disease, but atherosclerosis can also affect other arteries in the body [127]. A key step in the development of atherosclerosis is the accumulation of cholesterol into macrophages within the artery, leading to the formation of foam cells in response to pro-inflammatory activation of endothelial cells [17]. In addition, there is also strong evidence that oxysterols contribute to the pathogenesis of this disease. There are numerous oxysterol species that are elevated in atherosclerotic plagues and in the serum of patients with atherosclerosis including 27-HC, 25-HC, 7-KC, 7α-HC, 7β-HC, 5α, 6α-epoxide and 5β, 6β-epoxide as part of oxidised Low Density Lipoproteins (oxLDL) [18,128]. These oxLDL enriched with oxysterols can be taken up by cells in the vascular wall (including endothelial cells, smooth muscle cells and monocytes/macrophages) through scavenger receptors, such as CD36, scavenger receptor (SR) AI or AII and lectin-like oxLDL receptor 1 (LOX-1), having wide range effects [17,18,128]. In particular, 7-KC and 7β-HC are known for their pro-cytotoxic effects in atherosclerosis [10]. Exposure to 7-KC or 7β-HC induces apoptosis of endothelial cells associated with early lipid accumulation and lysosomal membrane permeabilisation, resulting in increased cellular oxidative stress and mitochondrial damage [129]. Another study illustrated that oxLDL or 7-KC can induce ER stress in endothelial cells detected by increased expression of phosphorylated IRE1 and eIF2α and persistent ER stress (indicated by expression of CHOP) leads to apoptosis [130]. In a similar manner, 7-KC was shown to induce apoptosis of aortic smooth muscle cells [131]. In terms of immune regulation, oxysterols amplify the production of inflammatory cytokines by atherosclerotic macrophages including the production of IL6, IL8, IL1β, and TNFα [132]. In the case of 27-HC, atherosclerosis is promoted through multiple mechanisms including acting on the ERα. Deletion of Cyp7b1 leading to increased 27-HC levels or direct injection of 27-HC promoted atherosclerosis in mice due to multiple proinflammatory effects mediated through attenuating ERα receptor function on endothelial cells and macrophages [92]. 25-HC also plays a role in atherosclerosis due to its proinflammatory function in macrophages. Following the deletion of ATF3 in mice, there was increased expression of Ch25h in macrophages and the elevated 25-HC production led to increased severity of atherosclerosis in these mice [133].

When Ch25h was initially cloned, is was found to have the highest expression in lung and this is likely due to high levels of expression in alveolar macrophages [134] (Immgen.org). Similarly, the human lung also has high levels of 27-HC produced by alveolar macrophages [135]. The high levels of oxysterols in the lung suggest that they have a role in lung homeostasis. 25-HC production in alveolar macrophages is required for phagocytosis of apoptotic cells and the resolution of lung inflammation [136]. However, in the context of chronic lung inflammation, oxysterols may have detrimental effects; 25-HC and 27-HC concentrations are increased in the lungs of patients with chronic obstructive pulmonary disease (COPD) and may have a role in neutrophilic inflammation [137,138]. Additionally, 25-HC can act as an amplifier of inflammatory signalling in the airway and is implicated in tissue damage due to increased production of inflammatory molecules such as IL6 [139,140]. Ch25h and Cyp7b1 expression are upregulated in airway epithelial cells in cigarette smoke-exposed mice and in COPD patients, suggesting increased levels of 7α-25-HC. It was illustrated that the 7α-25-HC and GPR183 migratory axis was critical in the positioning of B cells in inducible bronchus-associated lymphoid tissue (iBALT) formation and emphysema development in cigarette smoke-induced COPD, thus facilitating disease progression. Interestingly, in GPR183-deficient mice or following administration of Cyp7b1 inhibitor, clotrimazole, mice are protected against iBALT formation and cigarette smoke-induced COPD with less activated B cells [141].

Oxysterols are implicated in inflammatory bowel disease (IBD). In the colon of mice, the most abundant oxysterol is 4β-HC followed by 5α,6α-epoxycholesterol, with a range of other oxysterols present in smaller amounts [142]. Using multiple mouse models of colitis, both 4β-HC and 25-HC were consistently elevated in the colon. In biopsies from patients with Crohns Disease and Ulcerative colitis, 25-HC and 7α-HC were elevated but not levels of 4β-HC [142]. The strongest evidence linking oxysterols to inflammatory colon disease is for 7α-25-HC, which is a ligand for the cell surface receptor GPR183. 7α-25-HC is found to be elevated in both the colon and livers of mice in colitis models [142,143]. 7α-25-HC is important in the formation of lymphoid structures in the colon in a GPR183-dependent manner [143]. One source of the 7α-25-HC are fibroblastic stromal cells that express high levels of Ch25h and Cyp7b1 [81]. The mechanism involves the recruitment the GPR183-expressing innate lymphocyte subset ILC3, an immune subset with important roles in gut immunology, and in doing so facilitates the formation of these gut lymphoid structures and contributes to colon inflammation [81,143]. Mice deficient for GPR183 expression or 7α-25-HC production show reduced formation of these colonic lymphoid structures [81,143]. Therefore, the 7α-25-HC/GPR183 migratory axis is an important factor in gut immune homeostasis and for inflammatory processes in the colon.

Cholesterol metabolism and various oxysterol species are linked to the development of Alzheimers disease (AD). In particular, elevated 24-HC and 27-HC have been described in AD and are thought to play multiple roles in disease development [144]. Ch25h expression is increased in the brains of Alzheimers patients and in the brains of mice using Alzheimers mouse models and the 25-HC generated potentiates IL1β production from microglia [145]. Apart from AD, 25-HC is also elevated in the neurodegenerative disorder X-linked adrenoleukodystrophy and contributes to inflammation through Nlrp3 inflammasome activation [146].

### 5.4. Infection

For the most part, oxysterols play a positive role in the context of infection. Ch25h is an interferon-inducible gene and 25-HC has some well described protective roles in viral infection. 25-HC inhibits viral entry into cells, interferes with viral genome replication and with viral replication and assembly (see [1] for review). 25-HC has been shown to be broadly anti-viral and inhibits the infection of numerous viruses including HCV, HIV, Zika virus and Reovirus [55,147,148,149]. Macrophage-derived 25-HC also inhibits the infection of intracellular bacteria such as *Listeria monocytogenes* through interfering with cellular penetration of this bacteria [35]. In contrast, in the context of parasitic infection, oxysterols can have a negative impact. For instance, LXR signalling appears detrimental for leichmania infection in mice as LXR-deficient mice are protected from this pathogen due to enhanced macrophage-mediated killing that was associated with increased nitric oxide production [150]. Parasitic production of oxysterols can also have a role in certain diseases. Parasites such as the liver flukes *Opisthorchis viverrini* and *Opisthorchis felineus* generate carcinogenic oxysterols that can form DNA adducts in host cells leading to chromosomal abnormalities and a cancer of the bile duct called cholangiocarcinoma [151,152,153].

### 5.5. Endometriosis

Endometriosis is a condition where the tissue that normally grows within the uterus, the endometrium, can be found growing outside the uterus in various areas within the peritoneum, most commonly affecting the ovaries, fallopian tubes and the bowel. Once formed, the ectopic endometrial tissue becomes subject to immune surveillance leading to chronic inflammation. Patients with endometriosis are found to have dysregulated immune functions within the peritoneum, including increased regulatory T cells and macrophages [154]. Notably, the function of NK cells, the cytotoxic cells that should be killing the ectopic endometrial cells, is severely impaired [155,156,157]. It has been proposed that TGFβ produced by macrophages in the peritoneum causes this NK cell dysfunction [158,159]. However, there is also evidence that the inhibitory factor is not a protein as the ability of peritoneal fluid to inhibit NK cells function was not blocked by charcoal treatment [160]. It is also interesting to note that elevated levels of 25-HC are described in both the peritoneum and the plasma of patients with endometriosis [6]. Given that 25-HC can potently inhibit NK cell metabolism and function it is possible that this represents a contributing factor to NK cell dysfunction in this inflammatory condition.

### 5.6. Autoimmunity

Rheumatoid arthritis (RA) is an autoimmune disease characterised by chronic inflammation and destruction of cartilage and bones in the joint. Key immune contributors of RA are macrophages as they produce a number of pro-inflammatory mediators such as IL1β and TNFα, which are abundant in the synovial tissues and fluid in RA patients. Furthermore, patients with active RA show altered lipid metabolism, suggesting a strong integration of metabolism and inflammation [161]. Given the anti-inflammatory effects of LXR, it was proposed that LXR may be a possible therapeutic target for RA. Indeed, in a murine collagen-induced arthritis model, mice given a synthetic LXR agonist had reduced arthritis severity and decreased expression of inflammatory molecules including TNFα, IL1β, iNOS and COX2 [162]. Additionally, administration of a LXR agonist decreased serum levels of BAFF (B cell activating factor also known as BLyS, B lymphocyte stimulator), IFNγ and TGFβ. B cells and autoantibodies play a critical role in RA pathogenesis and there are increased serum levels of BAFF in RA patients compared to healthy controls. It was demonstrated that LXR activation in B lymphocytes leads to downregulation of both basal BAFF and IFN-γ/TGF-β-induced BAFF production [163]. Moreover, there is evidence that LXR activation can alter the balance of T cell differentiation, attenuating Th17 and Th1 differentiation while inducing the generation of Tregs [42,43,44]. Synovial tissue biopsies from patients that are at risk of developing rheumatoid arthritis were obtained and high risk individuals who progressed to inflammatory arthritis displayed increased levels of 25HC-synthesising enzyme Ch25h, while those with low levels of Ch25h remained arthritis free. Blockade of cholesterol biosynthesis in human CD4 T cells with the SREBP inhibitor 25-HC results in reduced levels of cMaf, a key regulator of IL10 expression, and consequently inhibits IL10 production while increasing IFNγ production. However, the mechanism of how altering cholesterol biosynthesis blocks IL10 and cMaf is unclear [164].

Multiple sclerosis is the most common autoimmune disease and is characterised by the infiltration of immune cells and demyelination of the central nervous system, leading to neuronal damage. Abnormal oxysterol levels are seen in MS patients and are implicated to be potential biomarkers of multiple sclerosis (MS), however the underlying mechanism is unclear. Of note, plasma levels of 24-HC, 27-HC and cerebrospinal fluid levels of 25-HC were significantly lower in MS patients compared to healthy controls [165,166]. The immunoregulatory roles of oxysterols on immune cells and how they relate to MS are emerging. For instance, in the murine model of EAE, mice given the cholesterol-lowering drug atorvastatin or a pharmacological LXR agonist had significantly reduced disease severity and overall inflammation [167,168]. It was previously reported that in vivo administration of a LXR agonist decreases Th17 polarisation and IL17A, IFNγ and IL23 production. Since Th17 largely contributes to EAE development, EAE clinical scores were also reduced when given LXR agonist [43]. In contrast, mice deficient for LXR had a higher frequency of inflammatory T cells producing more IL17 [41]. Other studies have suggested that 7α-25HC may play a role in trafficking inflammatory immune cells to the central nervous system as increased levels of 7α-25HC were observed in the brainstem of mice with EAE [169]. There is evidence to suggest that 7α-25HC produced by monocyte-derived DCs and microglia promotes the trafficking of encephalitogenic CD44+ CD4+ T cells that express GPR183 [170,171]. Two critical proinflammtaory cytokines, IL23 and IL1β, were required for the maintenance of GPR183 expression on these T cells [171]. Furthermore, Ch25h-deficient mice, which cannot generate 7α-25HC, have less severe EAE disease scores without showing any alteration in overall T cell and B cell responses. However, these mice do show altered trafficking of T cells, with reduced infiltration of IL17-producing CD44+CD4+ T cells in the CNS but increased numbers in the draining LN. It is important to note that Ch25h-deficient mice do not just reflect the role of Ch25h in producing 7α-25HC because these mice also are impaired for 25-HC. However, GPR183-deficient mice have a similar phenotype to Ch25h-deficient mice, arguing for key role for the 7α-25HC–GPR183 interaction in the development of this disease [170]. Interestingly, human memory CD4 T cells from MS patients treated with a common MS drug, natalizumab, display an increased expression of GPR183 and migration [172].

## 6. Conclusions and Future Perspectives

It is clear that there are a multitude of ways that oxysterols, these important cholesterol metabolites, can affect immune cells and immune responses (other review articles on this topic [71,173,174]). The effects that oxysterols have on the various immune subsets can be very distinct and both pro- and anti-inflammatory. In addition, these effects can be exerted intracellularly where the oxysterol species was generated or upon other cells in the local microenvironment and even more globally. Indeed, in various pathological situations, elevated systemic oxysterols have been observed. However, there is still much to be learned of the immunoregulatory properties of oxysterols. For instance, it will be interesting to further understand whether oxysterols have a role in intercellular communication between cells in a given microenvironment. Certainly, there are situations where immune cells that generate oxysterols are found to cluster with immune cells that are particularly sensitive to the effects of these oxysterols. A good example of this are NK–DC interactions, such as cDC1, which express Cyp27a1, and NK cells, which require SREBP for optimal function, are found closely associated in the spleens of MCMV-infected mice [175]. Is 27-HC limiting the NK cell response under these circumstances? Precisely understanding such communication events will require the generation of novel transgenic mouse models where oxysterol-generating enzymes, such as Cyp27a1, are deleted in specific cell types; in cDC1 for this given example. There are also several questions remaining regarding the role of oxysterols in the pathogenesis of various diseases. For instance, an interesting question is whether the elevated levels of oxysterols in patients with endometriosis contributes to NK cell dysfunction and disease progression. We think new and exciting immunoregulatory roles for oxysterols are likely to emerge in the future.

## Figures and Tables

**Figure 1 metabolites-10-00384-f001:**
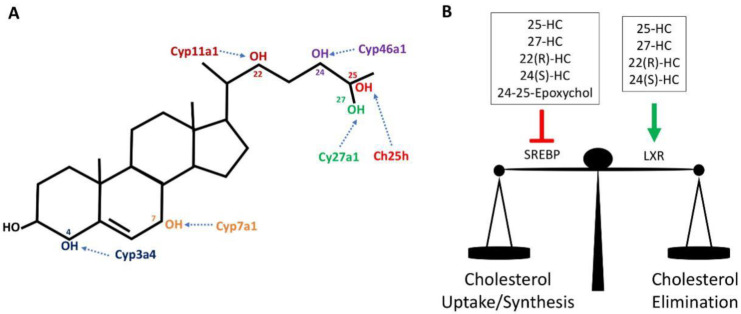
Oxysterols and the regulation of cholesterol homeostasis. (**A**) Structure of the 27-carbon cholesterol molecule, highlighting some of the carbon positions where enzymatic oxidation can occur, and the oxidizing enzymes involved. (**B**) The oxysterol species shown act to control cholesterol homeostasis through the regulation of two key transcriptional regulators, the transcription factor Sterol regulatory element binding proteins (SREBP) and the nuclear receptor Liver-X-receptor (LXR). Inhibition of SREBP inhibits de novo cholesterol synthesis and cholesterol uptake while activation of LXR promotes cholesterol efflux from the cell.

**Figure 2 metabolites-10-00384-f002:**
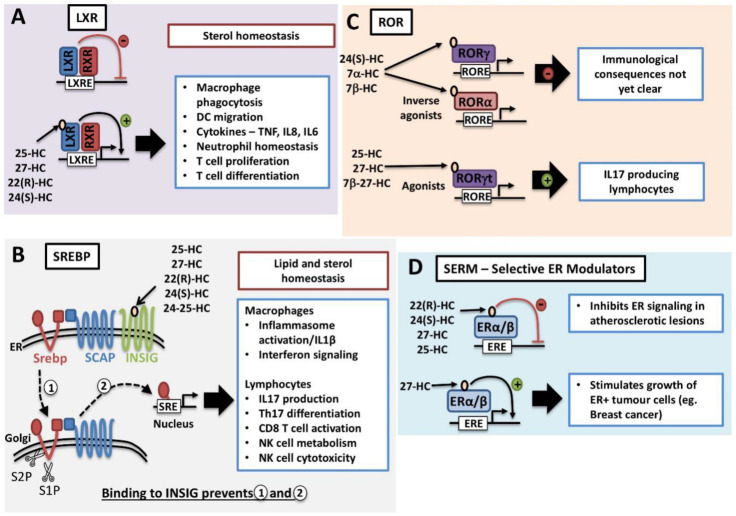
Oxysterols mediate transcriptional immune regulation. (**A**) LXR are nuclear receptors that form a permissive heterodimer with RXR and bind to LXR response elements (LXRE). In the absence of ligand, LXR/RXR act as transcriptional repressors. Oxysterol ligands, including 25-HC, 27-HC, 22(R)-HC, 24(S)-HC, bind to LXR and convert the LXR/RXR dimer into a transcriptional activator. Oxysterol regulation of LXR affects the function of diverse immune cells including macrophages, dendritic cells and lymphocytes. (**B**) Oxysterols inhibit the activation of SREBP transcription factors through binding to the ER anchoring protein INSIG, promoting its interaction with SCAP and thus preventing the trafficking of SREBP/SCAP to the Golgi (1) and the cleavage by site-1 and site-2 proteases (S1P and S2P). As a result, SREBP is not processed into the mature transcription factor (2). 25-HC, 27-HC, 22(R)-HC, 24(S)-HC and 24-25-Epoxycholesterol have all been shown to bind with high affinity to INSIG and promote the interaction with SCAP. Oxysterol inhibition of SREBP is important in the control of immune cells including macrophages and lymphocytes. (**C**) Certain oxysterol species can also act as ligands for α and γ isoforms of Retinoic acid receptor-related orphan receptors (ROR). 24(S)-HC, 7α-HC and 7β-HC act as inverse agonists for both RORα and for RORγ to inhibit ROR-dependent transcription. The immunological consequences of this inverse agonism are not yet clear. 25-HC, 27-HC and 7β, 27-HC act as agonists for RORγt and promote the expression of RORγt gene targets including the expression of IL17 in certain lymphocyte populations. (**D**) Oxysterols species can act as selective estrogen receptor (ER) modulators or SERMs. In atherosclerotic lesions 25-HC, 27-HC, 22(R)-HC, 24(S)-HC act to inhibit ERα/β-mediated signalling. In ER+ tumour cells, 27-HC stimulates signalling through ERα/β. SRE, sterol response element; LXRE, LXR response element; RORE, ROR response element; ERE, estrogen response element.

**Figure 3 metabolites-10-00384-f003:**
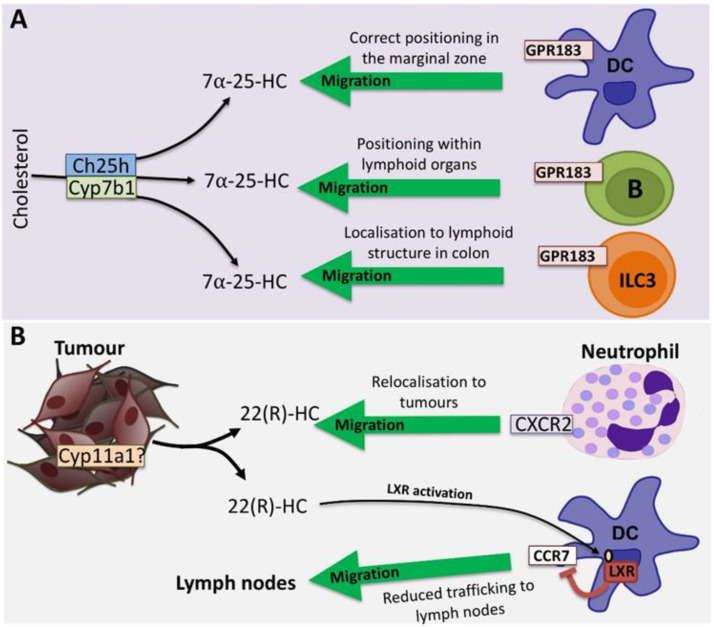
Oxysterols and chemotaxis (**A**) 7α,25-HC is an oxysterol that is generated from cholesterol by the sequential actions of the enzymes Ch25h and Cyp7b1, which acts as a direct ligand for the cell surface receptor GPR183. 7α,25-HC acts as a chemoattractant for cells expressing GPR183, which include B cells, dendritic cells (DC) and type 3 innate lymphoid cells (ILC3). 7α,25-HC-mediated chemotaxis is important for the correct positioning of B cells and DC within lymphoid organs and the homing of ILC3 to lymphoid structures within the gut. (**B**) Tumour-derived 22(R)-HC, most likely generated by Cyp11a1, can act as a chemoattractant for CXCR2 expressing tumour-promoting neutrophils. In addition, 22(R)-HC can act through the LXR in DC to repress the expression of CCR7 and in doing so inhibit CCR7-mediated migration of DC into lymph nodes.

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
