# Peer review of "Diverse Immunoregulatory Roles of Oxysterols—The Oxidized Cholesterol Metabolites"

_metabolites, 2020, doi:10.3390/metabo10100384_

Round 1
Reviewer 1 Report
This is a review on regulation, by oxysterols, of immune/inflammatory processes and their role pathological immune regulation. The manuscript is clearly written and comprises a wide range of mechanisms and diseases. While informative, there is plenty of room for improvement in several important aspects.
- While this review is on oxysterols, the focus seems to be largely on side-chain oxysterols. However, other oxysterols, including those with oxidized groups at the 7th carbon play very important roles in disease. For example, they are critical in induction of ER stress during atherosclerosis development, a process that, among others, increases inflammation. The role of these oxysterols should be expanded.
- As written, the oxysterol-ROR regulation seems misleading. Oxysterols are primarily depicted as ROR activators, yet there is evidence that some oxysterols (including 7-oxygeneted oxysterols and others) bind RORs and repress their transcriptional activity. In addition, it would be important to include information on the interplay between LXRs and RORs.
- An important aspect that is missing is on the regulation of endogenous oxysterols beyond some information on regulation of synthesis. A common pitfall of studies on oxysterols is that many of them have been performed using oxysterols at concentrations and in forms that do not necessarily represent a physiological scenario. Thus, in order to understand the function of oxysterols in immunology is critical to understand how endogenous and exogenous oxysteols are handled, trafficked, stored or secreted, etc.
Reviewer 2 Report
Title suggestion: "Diverse immunoregulatory roles of oxyterols - the oxidized cholesterol metabolites"
or simply "Diverse immunoregulatory roles of oxyterols"
Section 1- Introduction (needs some grammar edits):
- Missing citations on the first sentences. Citation 1 is for athero and oxyterols. Please add more citations to support the statements being made.
- The authors make some strong statements that need to be carefully reviewed, and expanded not to cause confusion. For instance, "elevated overall cholesterol in our bodies is bad for us and leads to atherosclerosis and heart disease". Please explain more. This is a strong statement and confusing interpretation can come from it.
Section 2
- Suggestion to change the title of this section to format "Oxidized Cholesterol - Oxysterols"
- Please mention how oxysterols are transported in circulation
Section 3
- Please make sure grammar is being edit. Some run on sentences are appearing throughout the manuscript.
- Missing citation to how the elevated oxysterols inhibit cholesterol uptake and synthesis (last sentence of first paragraph). Is the mechanism known? If so please describe here.
- Section 3.1: please mention what figure (a, b, c, or d) when referencing it in the paragraph.
- LXR beta is ubiquitously expressed in all other organs? If so, please add what reference this is mentioned.
- It is a bit difficult to follow the paragraph and figure 2, please detail more or change the order of the figures within figure 2 to make things flow easier.
- Missing citation after statements. Please add those.
- Section 3.2: first sentence is a bit confusing, please reword it.
- Please reference which Figure 2 the text is about.
- Suggestion: add the different oxyterols mentioned in the text to the figure 2a
Section 4
- Please chance the title of this section, the word control refers to regulation? Also consider another title for section 3. If oxyterols are "controlling" by binding to receptors or via paracrine fashion, then it seems like they regulate the immune system or its function. Is there a know threshold of oxysterols or any amount would be clinically relevant to regulate the immune system? Please expand on this topic.
- Section 4.1.1: on examples mentioned and work cited, were they testing oxyterols as ligands to LXR? If so please mention that in paragraph. Because on section 3.1 the authors mention that certain oxyterols species are natural ligands to LXRs, that is why this raises a question. Suggestion to also organize the line of thought in terms of actions of oxysterols in immune system. It seems like there is a lot of possibilities, but it would be appreciated if the authors organized the idea a bit more.
- Section 4.1.2: now the authors talk about tumor derived oxyterols? What is that? Please explain the difference. This concepts has not been introduced before and it is a bit confusing to the reader. Again the title says migration, but migration of what? It seem like only dendritic cells. Please fix.
- Section 4.2.1: Seems like it is only talking about macrophages and not other myeloid cells
- Section 4.3: please mention what part of figure 2 this is about.
- suggest mentioned what are the intermediate in cholesterol biosynthesis pathway mentioned in the last sentence that are ligands to ROR, are they involved with oxyterols? Can you elaborate a bit more in the relationship of these molecules if any?
- Section 4.4.1:again please mention what part of figure 3. Figure 3A is a bit confusing, the green arrows are not making much sense and the text doesn't seem to explain that very thoroughly
- Section 4.4.2: There is no context given to this scenario, please expand a bit come or contextualize the role of these other oxyterols species. Would it be in cancer? Again please mention what part of figure 3 the text is referring.
- Section 4.5: this is an excellent well structured section and paragraph which would be very helpful if the rest of the manuscript was contextualized in such manner. I would suggest modifying the title of this section because you are mentioning athero and the mechanism here doesn't seem to be estrogen receptor mediated, if so please add that to the athero part.
Section 5
- Section 5.1: nicely written and very well organized section. On second paragraph please distinguish the positive and negative impacts of oxyterols in tumor cells.
- Section 5.2: Is it known why obese males don’t have elevated levels of oxyterols? Are there any data, from animal studies regarding crown like structures in adipose tissues of animals with increased oxyterols? The anti-inflammatory effects of oxyterols in NAFLD model needs to be described a bit deeper. Is it known why this effect is being observed?
- Section 5.3: please elaborate more about role of oxyterols in colon diseases, the paragraph is very vague.
- Section 5.4: Are there any data regarding role of oxyterols in other viral infection, possibly corona virus (not the new corona virus of course since there’s a lot to be studied about it)?
- Section 5.5: great how the authors start the section describing the disease and then presents the role of oxyterols in the pathology, and finally concludes the thought. Doing this throughout the manuscript would certainly enhance the reading and understanding of this rich and highly needed review.
Reviewer 3 Report
This is a well-written review on an interesting topic in the field of immunometabolism. The following points need to be addressed.
(1) Relevant previous review articles on the role of oxysterols in immune function should be cited:
*Spann et al, Nat Immunol 14:893, 2013
*Cyster et al, Nat Rev Immunol 14:731, 2014
*Fessler et al, Trends Immunol 37:819, 2016
*Willinger, J Intern Med 10:2010, 2019 (Relevant to Section 5.3, Paragraph 449-453).
(2) The following original studies should be cited/discussed:
*Pereira et al, Nature 460:1122, 2009 should be cited regarding the role of GPR183 in B cell positioning in the spleen (line 283-285).
*Section 4.4.1: GPR183 is required for the proper positioning of ILC3s in the mesenteric lymph node. This information should be added and the two relevant studies cited (Chu et al, Cell Reports 23:3750, 2018; Melo-Gonzalez et al, 216:728, 2019).
*Section 4.4.1 (line 298-300): GPR183 promotes intestinal immunity against bacterial infection (Chu et al, Cell Reports 23:3750, 2018). This should be added.
*Section 5.3: GPR183 and oxysterols play a role in the formation of inducible bronchus-associated tissue, which contributes to the immunopathogenesis of COPD (Jia et al, EMBO Mol Med 10:e8349, 2018). This should be added.
Author Response
Please see attachement

Reviewer 4 Report
The manuscript written by Choi and David titled, “Diverse immunoregulatory roles for the oxidized cholesterol metabolites oxysterols,” provides an important knowledge of how oxysterols can affect the immune responses which could be anti or pro inflammatory in diverse immunological settings. The manuscript gives an important insight about studying the intra or intercellular role of these Oxyserols on local microenvironment. Overall, the manuscript is well written and should be accepted for publication. However, the manuscript should be thoroughly checked for common typos as
-In line 28. “Cholesterol can obtained” should be “Cholesterol can be obtained”.
-Line 220- catabolic metabolic. It should be either Catabolic or metabolic. Metabolism comprises of catabolic and anabolic processes.
Author Response
Thank you for these positive comments. We have checked carefully for typos as recommended.
Round 2
Reviewer 1 Report
N/A